# Customer Healthcare Complaints in Brazil Are Seldom about Medical Errors

**DOI:** 10.3390/ijerph21070887

**Published:** 2024-07-08

**Authors:** Arnaldo Ryngelblum, Marko Šostar, Berislav Andrlić

**Affiliations:** 1Graduate Program in Administration, Universidade Paulista, São Paulo 04026-002, Brazil; 2Faculty of Tourism and Rural Development in Pozega, Josip Juraj University in Osijek, Vukovarska 17, 31000 Pozega, Croatia; msostar@ftrr.hr (M.Š.); bandrlic@ftrr.hr (B.A.)

**Keywords:** complaints, comprehensive healthcare, institutional logics, consumer satisfaction

## Abstract

This study reviewed different country studies and noted that complaints in Brazil are more concentrated in complaints about being attended to and receiving access to services, rather than about clinical quality and safety issues. This paper explores the possible explanations for these differences based on the institutional logics theory and which logics actors privilege, and how they may play out in the healthcare field. To accomplish this undertaking, this study makes use of the healthcare complaint categorization developed by Reader and colleagues, which has been used by various studies. Next, a set of studies about healthcare complaints in different countries was examined to analyze the issues most common in the complaints and compare this information with the Brazilian data. This study identified three explanations why complaints about medical errors seldom occur. One group of studies highlights the hardships of local health systems. Another focuses on patient behavior. Finally, the third kind focuses on the issue of power to determine health orientation. The studies about a lack of resources do not directly explain why fewer complaints about clinical quality occur, thus helping to stress the management issues. Patient behavior studies indicate that patients may be afraid to point out medical errors or may be unaware of the procedures of how to do so, suggesting that family logic is left out of the decisions in the field. The third group of work highlights the prominence of the medical professional logic, both in terms of regulation and medical exercise.

## 1. Introduction

Studies about consumer complaints focus on different topics, such as complainants’ characteristics, dissatisfaction questions, and complaints about organizations’ responses to the issues raised [1].

Specifically, healthcare complaint questions were identified and subdivided into seven categories, expressed by three conceptually different domains [2]. These domains include safety and quality of care, management of healthcare organizations, and healthcare staff–patient relationships.

Studies on healthcare complaints carried out in countries such as the UK, the USA, Australia, New Zealand, Sweden, and others show that the safety and quality of treatment are significant aspects of all complaint manifestations [2,3,4,5,6,7], which differ from the Brazilian experience.

In Brazil, there are two complementary health systems. One that is public and universal (SUS, Sistema Único de Saúde) attends to the majority of the population but fails to provide care in many situations [8,9], and the other is a private system that includes health operators, insurers or mutual companies, and independent providers. The complaints can be addressed to providers, such as hospitals or laboratories; to the courts; to a medical council (CFM, Conselho Federal de Medicina and its regional branches); to regulatory agencies, one for private health (ANS, Agência Nacional de Saúde Suplementar), another responsible for medicine approvals and supervision (Anvisa, Agência Nacional de Vigilância Sanitária); an agency dedicated to consumer issues (Procon, Fundação de Proteção e Defesa do Consumidor); and complaints can also be addressed to consumer nongovernmental organizations. However, medical safety and quality treatment problems are handled only by providers, the CFM, or the courts. The other channels usually handle only management, relationship, or access issues.

However, hospitals do not provide detailed information on consumer complaints about medical errors and other issues [10,11]. Analogously, the CFM regional branches inform aggregate data only on ethical–professional lawsuits [12,13]. The topics that appear more frequently as claim demands and justifications to the courts are restrictions to obtain medicines, prostheses, exams, and hospitalization, according to a study commissioned by the National Justice Council (CNJ), an agency of the judiciary that is responsible for setting the basic guidelines to the courts about health judicialization in Brazil, given the strong increase in appeals, 130% between 2008 and 2017 [14]. However, the study, again, brings only aggregate figures, although they cover the entire country. The researchers experienced several problems completing the survey, from difficulties accessing the data to standardization issues. Nonetheless, the fact that the majority of claims to the courts involve issues other than safety and quality of care is a strong indication that the profile of the complaints in Brazil is different from that in other countries.

This study examines the similarities and differences between healthcare complaints in Brazil and different countries based on the literature and available data. It then explores the possible explanations for these differences based on the institutional logics that the actors privilege and how they may play out in the healthcare field.

To carry out this undertaking, we start by presenting the healthcare complaint categorization developed by Reader et al. [2], which has been used by various studies [3,15,16]. Next, we examine a set of studies about healthcare complaints in different countries to determine the issues that are most common in the complaints. We compare this information with the Brazilian data. The studies were intentionally selected to present the situation in different countries and compare with Brazilian studies about the issue. In the following section, we present institutional models to help analyze the behavior of field actors and the reasons for the outcomes shown.

## 2. Healthcare Complaint Categorization: Methods

Reader et al. [2] developed a complaint categorization based on 59 publications reporting more than 88 thousand complaints. They analyzed these complaints by developing different codes, which were combined in 29 subcategories of complaint issues. These subcategories were then grouped into seven categories that were additionally classified into three conceptually distinct domains.

Reader et al. [2] (p. 678) mention a complaint figure of more than 100,000 annually on hospital care in the NHS (National Health Service). The articles examined were restricted to those that reported primary quantitative data in English, mostly from the UK, the USA, and Australia, in which 86% of the complaint targets were medical staff [2] (p. 681). The healthcare environment complaints included mainly hospitals and emergencies.

In these studies, the complaint issue percentages examined are similarly divided between the domains. The safety and quality of clinical care issues represent 33.7% of the total value. The other domains include the management of healthcare organizations and healthcare staff–patient relationships.

A further study by two of these authors [15] revised some subcategories of the previous work and added the notion of severity to the complaints. Thus, the clinical domain was subdivided into quality and safety categories. The first deals with inadequacies in treating patients, while the second includes errors and deficiencies. The management domain comprises institutional and environmental issues, where the first comprises bureaucratic topics, access, billing, and supporting services for patients. The second has to do with facilities, admission, discharge timing, and delays. The relationship domain is subdivided into communication, listening, and patient rights categories. Communication involves inadequate, inconsistent, wrong information, and dialog. Listening concerns poor attitudes and a disregard of patient information. Rights are about abuse, confidentiality, or discrimination [15] (p. 941).

In the next section, we make use of Reader et al.’s categorization to examine what issues are most prominent in the complaints shown in the studies of different countries. When this specific categorization is not used, we try to identify its best approximation.

The studies included in this analysis were selected in the Scopus base, looking for healthcare complaints and dissatisfaction. We searched for quantitative studies of complaints that presented reasons for the complaints, following the classification from Reader et al. [2]. Brazilian studies were selected from the Scielo base.

## 3. Findings

### 3.1. Healthcare Complaining in Different Countries

Reader et al. [2] (p. 685) mention that “patients were found to complain almost equally on the domains of ‘clinical’, ‘management’ and ‘relationships’. Some institutional factors appear more specific to certain healthcare systems (e.g., finance and billing in the USA)”. The American experience has much to do with complaints directed toward doctors and medical entities, in addition to problems with costs and bureaucracy [17].

O’Dowd et al. [3] also use complaint categorization from Gillespie and Reader [15] to examine potential interventions to improve quality based on the complaints. The article explicitly mentions that in the Republic of Ireland and the UK, the more statistically frequent type of complaint is harm suffered by patients in the care wards, examination and diagnosis, and operations and procedures.

Sundler et al. [4] examined complaints in Sweden and described them under three themes: the right to available and accessible healthcare services, the right to good quality healthcare services, and respect for dignity and equality in healthcare.

In their study, Schnitzer et al. [7] examined complaints directed to the Federal Commissioner for Patient Issues in Germany, authorized by the 2004 healthcare reform act to handle patient concerns. The survey noted unjust policies, refusal or restriction of drugs, and refusal or restriction of nondrug treatment as the main issues raised, while complaints about the physician–patient relationship doubled during this period. These complaints were primarily lodged by statutory health insurance holders, who make up 88% of the population. Self-employed individuals and certain employees can opt for private health instead of statutory health insurance. In this study, the issue of unjust policies is hardly comparable to Reader et al.’s categories, but it suggests discontent with rules about using the services.

Mattarozzi et al. [16] carried out research on complaints about a hospital in Bologna, Italy, using the categorization taxonomy developed by Reader et al. [2]. According to their analysis of several complaints, the most common causes were the time taken to access treatment and the communication of issues. These are classified under the management category. Clinical or quality questions were fewer in number and were almost always associated with relationship issues.

Using a particular complaint categorization, Jiang et al. [18] identified several categories and subcategories that justified complaints toward a tertiary hospital in China during a 5-year period. Among these, complaints about treatment processes were more common than complaints about other processes. The doctor–patient relationship was also an important item.

Natowicz and Hiller [19] present a study about newborn screening programs in the USA and the redress of grievances. The researchers contacted those responsible for newborn screenings about medical conditions regarding reception protocols and the correction of reported problems in American states. The complaints listed included information failure, parental consent/refusal, communication of results, errors and exchanges of hospital samples, screening procedures, and laboratory errors.

A study by Nikodem et al. [20] indicates that trust in the Croatian healthcare system is increasing, but problems are still evident in the patient–doctor relationship, where there is not an adequate level of communication and trust. Karabatić et al. [21] reveals a high level of patient dissatisfaction in Croatia, highlighting the lack of sufficient bidirectional communication, where the patient does not receive enough information and the attitude toward them is not satisfactory. A study conducted by Mermolija [22] in Croatia revealed parents’ dissatisfaction with not being allowed to participate in their child’s care across different healthcare settings, especially noting that those with children suffering from acute illnesses were generally more dissatisfied with healthcare quality than the parents of children with chronic conditions. Finally, when examining the adoption of patient safety culture practices in hospital processes, there is evidence suggesting that public hospitals in Croatia exhibit relatively scant error reporting, a lack of accountability and feedback-driven improvements, a strict top–down decision-making hierarchy, and insular operations [23].

One possible conclusion to draw from the above studies is that the majority of the cases conform to Reader et al.’s domains; that is, they all have issues related to safety and quality, management, and relationship issues in similar proportions. There are a few exceptions, wherein in some of the cases, the finance item is more outstanding or has specific restrictions.

### 3.2. Healthcare Field and Its Influence on Complaints

Several studies sketch the motives present in complaint manifestations that may help explain the specific characteristics that each country experiences.

Sundler et al. [4] commented that although complaints may be important in the improvement of healthcare procedures, formal patient complaints are relatively rare because they feel that the medical perspective is generally privileged [5] and that complaints seldom contribute to improvements [6]. A Norwegian study also underlined the power of the medical system in relation to the issues raised by patients [24].

Gal and Doron’s [25] study examined the prevalence of informal complaints not made through designated channels on healthcare services in Israel and reported that only 25% of the participants reported a cause [17] and less than 10% actually filed a complaint. Schlesinger et al. [17] estimate that consumers will only express dissatisfaction if they expect it to be worth the effort.

Coyle [26] reviewed a number of studies that focused on formal complaints to the NHS and showed that the majority of complaints relating to hospital care are not about clinical matters and that complaints against doctors have to do with poor communication, failure to diagnose, and dissatisfaction with treatments. However, these studies indicate that the picture must be incomplete, given that cultural and psychosocial barriers prevent patients from complaining, such as the stigma associated with one’s behavior, the fear of being called a nuisance or a neurotic, the gratitude factor when balancing service questions against results, and the difficulty in verbalizing what patients feel.

Healy and Walton’s [27] article compares the structures of ombudsmen (health complaint commissioners) in England, New Zealand, and several Australian states. This function responds to complaints, but few cases evolve into investigations and prosecutions. Health regulation, on the other hand, is assumed by state and nonstate actors and not by a centralized authority, although the ombudsman can establish their own regulations and the capacity to regulate health providers [27] (p. 499), being able to investigate and, with the exception of England, prosecute providers.

McCreaddie et al.’s [28] article examines the rhetoric of complaints to the NHS, which largely focuses on treatment and care, specifically on communication and relationship aspects, rather than administrative issues. At the same time, the authors raise an important point that complaints may not be sustained because they are investigated by the same party against whom the complaint was made.

In summary, the studies mention that the reasons for fewer complaints than expected seems to be due to physicians’ capacity to privilege their points of view or to the structure erected to deal with these cases.

### 3.3. Healthcare Complaints in Brazil

This section reviews two kinds of materials, the first about studies related to healthcare complaint issues in Brazil and the second about complaint reports obtained from the several locales that receive these claims, which complement the picture relating to the theme.

Gomes and Amador [29] noted that the majority of administrative lawsuits against the federal SUS (the national health system) involve obtaining medicines, which is a right that patients have in the public system but are sometimes denied or the medicines are unavailable. A total of 45 studies were selected, but 11 were reviewed. Their interest was to identify the lawyers who represented the complainants, and whether they were public or private. The authors noted that lawsuits are concentrated within a reduced number of lawyers, doctors, and pharmaceutical companies. The main allegations involved health urgency, risk of death, and prescription by doctors.

Freitas et al. [30] reviewed the literature on health judicialization during the period 2004–2017. The majority of lawsuits involve obtaining medicines [31]. The lawsuits are then a solution to solve an emergency. Schulze and Neto [32] comment that although patients may have rights recognized, they may see refused demands forwarded through administrative processes. However, lawsuits may also be used when there is no legal provision for a service or a good, which is the case for medicines or medical procedures that are not yet approved. In this review, a few studies noted that a concentration of doctors and private lawyers led the suits. On the other hand, public defenders represent patients with few resources to hire a lawyer. Private health demands, in turn, experience denials of procedure coverage as the main complaint item, especially costlier demands, such as cancer treatments and treatments for cardiovascular diseases.

Carvalho et al. [33] conducted a complaint survey of two channels that do not usually receive grievances related to treatments and procedures. These are the government complaint site consumidor.gov.br and the regulatory agency ANS’s site. From May 2014 to May 2018, they observed a great concentration of complaints against a few large healthcare operators, namely companies that intermediate the provision of services in private healthcare. While consumidor.gov.br displayed only approximately 7000 claims, ANS had almost 400,000. In the first case, the majority had to do with undue billings, problems with the operators’ customer service, and with contracts, while in the second, the majority were about the coverage of treatments. According to the ANS [34], complaints about health plans and insurance operators have grown almost 200% between 2019 (before the pandemic) and the first ten months of 2023, from 363 to 973 per day on average. The main complaint has to do with the management of requests, followed by reimbursements and service timings, but without any mention of clinical outcomes.

In another study of a few of the northeastern states of Brazil, Silva Junior and Dias [35] confirmed the difficulties patients faced when accessing medicines and a lack of knowledge of their rights, which generated a low appeal to the courts. In the private health system, in São Paulo, a survey by the Associação Paulista de Medicina, the association of doctors, described that patients complained about denials of treatment coverage, waiting queues, delays in service, and early release in the public system. Problems in private healthcare include emergency care and delays in scheduling appointments and carrying out diagnostic tests [36]. The judiciary’s stance to respond, in most cases, positively to health claims has led to a large increase in health costs, which benefits most patients with greater resources.

Siqueira [37] reports the number of complaints registered with the ANS between 2011 and 2015 against three of the bigger healthcare operators in the field, as well as the claims made against them. Amil Assistência Médica Internacional faced 47,995 processes, and the reasons presented involved the management of health demands, contract suspension or termination, service provision by the affiliated network, coverage of procedures from the official list, and service period extent. The other two faced fewer claims but with similar motives.

Ouverney [38] tells us about patients who appeal to the public defender’s office, which is constrained to attend only to those with scarce resources. The author mentions that the majority of the lawsuits, administrative or judicial, are about medicines that were either unavailable or refused, while the difficulty or impossibility of scheduling surgeries comes next, in a much smaller percentage. The defender’s office in Rio de Janeiro registered 16,343 processes in 2011. Figueiredo [39] states that people are unaware of administrative procedures to have access to medicines, and that health professionals have difficulty getting to know the list of medicines available to the public. Another problem that the author calls attention to is that medicine supplies resulting from court decisions are acquired without licitation, which encourages fraud. Vasconcelos [40] sets out to analyze court decisions that include not only what she describes as the decision making of judges, but also several other offices, both in the judiciary and the public administration. Her study included four states with the highest numbers of lawsuits, confirming that the demand for medicines in the public system was the main problem.

The study by Insper [14], a private research organization, conducted on behalf of the CNJ, painted the picture of the judiciary’s role in the health sector in the country. Several difficulties are reported in the survey process, ranging from problems of access to standardization issues. The study revealed the following complaints: ‘health plans‘, ’insurance’, ‘health’, ‘medical-hospital treatment’, and ‘supply of medicines’ (p. 15), which suggested that the research did not delve into the specific demands and justifications of the lawsuits. The more frequent theme in the public system is the provision of medicines, and others more common in supplementary health are diet provision, input of materials, beds, and medical procedures. The work analyzes whether there is an asymmetry in the figures of judicial lawsuits due to the growing proportion of cases related to private health in relation to public defenders that handle cases of the public health system (p. 20).

Vieira Junior and Martins [41] analyzed elderly (older than 60 years) patients’ complaints against private health operators, verifying that between 2010 and 2012 they consisted of more than twice the complaints of the other groups. The study proposed that this age group suffers more access restrictions to health services because they generally imply higher costs. Thus, their entry into these plans is subject to marketing constraints, such as the demands of medical reports of previous conditions to be able to join, and in many cases, they require previous authorization from the administration to make use of services.

The online site Reclame Aqui (complain here) [42], which records any kind of complaint, features an excellent reputation for the prestigious hospital A. Einstein for answering all the claims posted and resolving several of these claims. Specifically, when browsing through complaints, it is difficult to find any complaint related to medical errors or poor clinical outcomes. The hospital edits a report [10] that states that patients report satisfaction with the treatments received in some of its main areas of expertise (cardiology, oncology, and neurology, among others). Another prestigious hospital, Sírio-Libanês, also features well [43]. The CNJ statistical site (2024) includes complaints about medical treatment that are usually less common than other issues, especially concerning the supply of medicines.

The above studies provide a picture of complaints that do not report clinical quality or safety problems. Nonetheless, there are studies that examine this situation. Delduque et al. [44] found 693 appeals to a higher court of the federal capital during the period 2002–2019 about medical errors. Bitencourt et al. [45] state 238 ethical procedures against doctors from 2000 to 2004 in the Medical Council of the state of Bahia. Fujita and Santos [46] reported approximately 724 claims about medical errors and other types of complaints, such as inadequate doctor–patient relationships and inappropriate behavior, registered at the Medical Council of the state of Goiás during 2000–2006. Mendonça and Custódio [47] mention that the São Paulo State Medical Council received and analyzed 376 claims about medical errors addressed to the judiciary between 2000 and 2004. A review by Silva et al. [48] of publications on medical error claims between 2013 and 2023 revealed figures in a similar range to those of the above studies. An article by Mendonça et al. [49] interviewed physicians to understand their perspective on making errors, which evidenced the fear of lawsuits, shame, and difficulty disclosing this kind of information to the patients.

Couto et al. [50] reported that adverse events occur in at least 4% of patients per year, which can mean more than 500,000 people [51]. In the period between June 2014 and June 2016, Metelski et al. [52] reported 63,933 adverse events in Brazil, but this figure is likely underestimated. Given the figures on adverse events, some of which may be due to medical errors, we could expect the number of lawsuits or complaints to be much greater.

In summary, most of the studies in this section mention the difficulty of obtaining medicines as the main cause of prosecution or complaint, predominantly in public health services, while procedure coverage is the main issue in private health. Using the categorization of Reader et al. [2] in the Brazilian complaint set, it is noticeable that the main complaints and claims have to do with the domain of management of healthcare organizations and some with healthcare staff–patient relationships. Moreover, the safety and quality of clinical care issues, reported as medical errors or ethical procedures against doctors, make up a very small number of clinical care issues compared to other issues.

### 3.4. Exploration of Complaint Differences between Brazil and Other Countries

Studies about healthcare complaints in different countries mention safety and quality, management, and relationships as relatively important issues. The Brazilian cases suggest that patients are more concerned about complaining about being attended to and receiving access to services that may be denied, even when the patient has a right. The weight of complaints about clinical quality and safety is proportionally much lower than that of other problems. However, as Couto et al. [50] and Metelski et al. [52] conjectured, this figure may be underestimated because patients do not file a complaint for several reasons.

Sundler et al. [4] described access theme issues mentioning a refusal to transport a person to an emergency service due to a mistaken evaluation of the person’s condition; in these cases, we understand that, under a better assessment, the service would be provided. In contrast, the Brazilian cases suggest denials in many situations irrespective of rights. In another case reported in the study, surgical cancellations were due to staffing issues, which indicates that once the surgeries were resolved, they would take place. In Brazilian studies, on the other hand, surgical denials are part of the usual refusal of procedure coverage. Finally, the study describes the quality theme where patients have been injured or suffered from poor-quality treatment. The dignity and equality theme is exemplified in relationship problems between patients and medical staff. In Brazil, as described above, lawsuits and claims against poor clinical treatment are rare.

In the Brazilian legal system, the duty to prove facts and rights follows the provisions of article 373 of the Code of Civil Procedure. Note that the author of the lawsuit is the one responsible for doing this and establishing their right in this way. It is possible to do so through expert, documentary, or testimonial evidence. In cases involving technical aspects, the evidence must be from an expert. Thus, the victim of a medical error will have to prove the fact (the medical error) and the damage (material, moral, aesthetic, psychological, existential, and other) to be able to claim compensation resulting from an error made by a doctor. The main difficulty lies in this process: doctors rarely point out mistakes made by their colleagues. Expert reports are almost always inconclusive. Alternatively, when there is a favorable report for the victim, the doctor accused of error obtains favorable reports that contradict the previous conclusion and raise doubts that benefit the accused.

One proposition that we may extract from the comparative cases is that the Brazilian patient faces greater difficulty accessing their health rights. Additionally, they are much more afraid or in disbelief that their actions can correct medical procedures or that they should try doctors in the courts. 

### 3.5. An Institutional Logics Approach for Complaint Behavior

The institutional logics approach understands that multiple logics are usually present in a field or context [53]. The field is usually understood as the reunion of all actors who play a role in a specific set of activities meant to achieve objectives and outcomes [54]. The field is composed of actors, organizations, and individuals who may have a more decisive and frequent role in the events taking place in the field and those that are less present and play a minor part.

The logics in the field are socially constructed patterns of tangible practices and cultural subjective elements, such as values, assumptions, and understandings, which guide the actors in the field by providing meaning and orientation to activities [55]. Institutional authors agree about the possibility of professional, market, state, democratic, community, family, and religious logics manifesting in the field [53,55,56]. Considering that each logic has directives about how to do things, they may indicate contradictory procedures at certain moments and, therefore, generate conflict about which directives to adopt. For instance, a professional logic concerned with appropriate procedures may conflict with a market orientation that can prioritize agility at the expense of professional care.

Complex fields are composed of numerous actors who exercise different activities in the field. Some of them participate in a service chain, such as in a client–supplier relationships; others act as regulators, as educators, as intermediaries, and so on. These actors are subjected to multiple field logics, but they tend to prioritize one depending on the specific issue under discussion. Field participants promote these logics to assert their own values and interests, which leads to a competition of logics regarding the definition of issues in the field [53,56].

The definition of how certain practices will be carried out and which structures will support them has also been the subject of several institutional studies [57,58]. To support the present study, it suffices to say that the logic guiding a specific issue may contain guiding aspects of different logics, which constitute the guiding logic for that issue, at a certain moment.

When we consider the healthcare complaint field, some of the main actors are the patients, their families, and their close acquaintances; the healthcare operators, health plans, and insurance companies; their providers, such as hospitals, clinics, laboratories, physicians, and others; medical suppliers; regulatory agencies; and the judiciary.

The procedures that may be used to complain and forward a demand are already in place, such as prosecutions, administrative processes, or directing the complaint to the hospital or operator ombudsman and broadcasting the issue to the media/social media. Currently, the underlying mechanisms (practice and structure) are available for disputing a problem in the field. Patients and families complain of obtaining redress, compensation, and explanations in relation to the problems they have endured. Doctors and hospitals may support, for instance, the claim for a medicine that has been refused by a public hospital.

These are generic mechanisms that allow complaint forwarding and redressing. However, the way these operations function in the various countries is different because the construction of complaint activity was historically different between them.

## 4. Examination of Reasons for Complaint Differences

The various studies indicate different conditions that may help to explain the aspects laid out in the proposition about the difference between the Brazilian experience and that of the other countries that were described.

The first kind of study focuses on the hardships of local health systems. One of them is that the Brazilian patient faces greater difficulty accessing their health rights. Schulze and Neto [32] support this proposition by saying that although patients may have rights recognized, they may see their demand refused. The main difficulty, which was repeated in every study, was obtaining the medicines needed for treatment. In other countries, the occurrence of such issues is proportionally less common.

This situation is produced by the public health service (SUS), which is supposed to provide medicines. Three conditions produce this kind of action: lack of resources (for instance, expensive new technology), supply delay [8], and fraud. The latter is expressed in previous studies by mentioning the concentration of demands on a reduced number of doctors and lawyers [29,30].

The logic that prescribes these actions in the SUS involves legislation; infrastructure; how to operate hospitals; how to contract services; who is eligible for treatment, which in this case is universal; and other aspects. The logic combines elements from the medical professional logic with elements from the state logic, which has to care for its citizens, organizations, and society as a whole, together with a family logic, which is concerned with the wellbeing of its members. However, when there is a lack of resources to acquire medicines, patients are occasionally left without them, which speaks against the public system, harming its legitimacy. This phenomenon leads people to look for alternatives to satisfy their needs, something that can bring about the production of another logic with a different orientation. Market research shows this tendency, whereby Brazilians wish to join a health plan, listed as their third most important desired item [59].

The refusal to provide medicines due to suspicion of fraud by physicians and lawyers is a measure that runs against the prevailing logic directives, which may be denied by the courts and is an effort by the SUS, following a state logic, to try to prevent being defrauded. However, as Figueiredo [39] states, medicine supplies resulting from court decisions are, in general, acquired without licitation, which encourages fraud. The studies about a lack of resources do not directly explain why fewer complaints about clinical quality occurred; rather, they stress the inclination for complaints about management issues, this way pointing to the few ones about quality.

The second kind of study focuses on patient behavior. Thus, a proposition provided by the analysis of the different experiences is that the Brazilian patient is afraid to prosecute doctors or does not believe their actions will result in any improvement in the field. This is apparent when examining the figures of adverse medical events, which are thought to be underestimated [52], and those of lawsuits related to medical errors, which constitute a small portion of that figure. These findings lead us to assume that healthcare complaints are seldom about medical errors, that is, those regarding clinical quality and safety.

Different explanations appear in these works. Figueiredo [39] confirmed that patients are unaware of administrative procedures to have access to medicines, which suggests that they would also face the same problems when complaining about clinical outcomes. Gillespie and Reader [15] proposed that it may be unclear to patients how to complain, believing it to be ineffective or fearing negative consequences from their attitude [26]. Silva Junior and Dias [35] underlined their lack of knowledge of rights, which benefits a minority with greater resources [14,36].

According to other perspectives, only a small percentage of patients complain [25] about the errors, and this is the case only if any improvement is expected from the action [17]. We also know that the legal system established in the Civil Code does not encourage complaints; rather, it discourages them. Still based on explanations related to patient behavior, Coyle [26] speaks of cultural fears as seen as a nuisance or a neurotic manifestation and the gratitude stance that tries to balance pros and cons.

Under these circumstances, we must understand that patients, families, and consumer associations do not have a say in the way the field operates and that physicians, medical organizations, regulatory agencies, and the Congress deliberate all alone. This means that the family logic is kept out of the field’s decision, and this can be partly explained by the cultural belief patients hold in relation to their relationship with doctors. Experience everywhere tells us there is some truth in what goes on in the field.

In the third kind of work, the medical perspective is seen as exempt from criticism because of its power in the system [4,5,6,24]. McCreaddie et al. [28] also follow the power explanation, arguing that those that were complained about are the ones to investigate, which leads to the disposal of the complaint. On the other hand, Healy and Walton’s [27] view is that the majority of cases are resolved amicably by the ombudsman in England, New Zealand and Australia.

The Brazilian experience confirms the explanation about medical power; but since there is a variety of possible complaint venues, it is suggested that patients have to dispute the logic prescription to complain with other actors. However, the analysis of the power perspective is similar to that of previous studies in that powerful actors decide among them the logic orienting the field. This is usually the case here, given that doctors are at the same time practitioners and managers of the organizations running the field.

The smaller weight given to quality and safety issues tends to underline management and relationship issues in local complaints [29,30,31,33,34]. In this regard, some of the management systems used in this process should be proposed—CRM (Customer Relationship Management) [60].

Finally, complaining about healthcare provision indicates that dissatisfied patients may seek to evade the prevailing logic about how to complain and obtain a satisfactory answer or redress by appealing to the courts, for instance. There are also those cases in which the patient does not challenge providers or operators because they are afraid or does not know how to proceed. If we accept the explanations presented by the studies, including those of actors with power, then we must accept that there are different behaviors in the field. Patients may either follow logic prescriptions or evade them in some of its aspects.

The hardship explanation has to do with the behavior of providers or intermediaries, such as health operators. The reasons given about lack of resources, delays, and frauds are also evasions of the prescribed activities, which may be the fault of third parties but nonetheless do not follow the practices that are prescribed.

## 5. Conclusions

The present study concentrated on the dissatisfaction aspects related to healthcare services reported in various studies about specific experiences in different countries. Important distinctions in complaint behavior between patients in several countries and in Brazil were noted.

While in the studies about complaints in other countries, medical quality and safety issues feature approximately the same proportion as those relating to management and relationship issues, in Brazil, the participation in complaints about medical errors is disproportionate to the other two. The Brazilian cases suggest that patients are more concerned about complaining about being attended to and receiving services than about clinical quality.

This study used the institutional logic perspective to understand why this is so. Basic mechanisms for forwarding complaints and attending to them exist all around, but with important differences due to the construction process conducted in each context.

Thus, based on the studies reviewed, we were able to identify three distinct explanations for why complaints about medical errors seldom occurred. One group of studies highlights the hardships of local health systems. Another focuses on patient behavior. Finally, the third kind focuses on the issue of power to determine health orientation.

These explanations focused on an interesting aspect of behavior in a field. It showed how different actors behave very differently; in this field, one case involved patients who complained and even appealed to the courts, compared to those who preferred to let the dissatisfaction pass or did not know how to forward a complaint. Another example are those providers who face or do not know how to deal with a lack of resources, a supply delay, or frauds, including those who would rather turn a blind eye. The different actors behave very differently in the health field, opting for different actions that may correspond to the prevailing logic or evade it, risking a penalty or giving up altogether.

This research has limitations that involve a reliance on studies that are only able to suggest some trend. This work was based on a limited number of studies on healthcare complaints, which may be expanded upon in future work so that the knowledge about the reasons for dissatisfaction with complaints may improve. An alternative is to research the actors directly in the field.

## Data Availability

Not applicable.

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
