# Peer review of "Customer Healthcare Complaints in Brazil Are Seldom about Medical Errors"

_ijerph, 2024, doi:10.3390/ijerph21070887_

Round 1

Reviewer 1 Report

Comments and Suggestions for Authors

Dear Authors,

Thank you for the opportunity to review your paper.

Overall, while your manuscript shows extensive effort, it is quite long and heavy to read. It offers minimal new insights, and an empirical study or systematic review would have been more beneficial. The current manuscript reads more like a discussion article or narrative review and does not add enough to the existing literature in this field.

Author Response

Dear Authors,

Thank you for the opportunity to review your paper.

Overall, while your manuscript shows extensive effort, it is quite long and heavy to read. It offers minimal new insights, and an empirical study or systematic review would have been more beneficial. The current manuscript reads more like a discussion article or narrative review and does not add enough to the existing literature in this field.

Dear Reviewer

You are right about doing an empirical study. Still, we made a selection of foreign quantitative studies in the Scopus base and in the Scielo base, for the Brazilian, which indicated the striking difference between the distribution of the kinds of complaints, which asked for an explanation.

Submission Date

16 May 2024

Date of this review

27 May 2024 15:04:43

Reviewer 2 Report

Comments and Suggestions for Authors

This manuscript is very well written, yet it needs some minor changes to be done prior to publishing.

The abstract should contain an aim as well as a conclusion.

The authors should omit referencing in the abstract section.

Also, at the end of the introduction, the authors should state the aim of this research.

The keywords should be according to the PubMed Mesh subheadings.

The authors should introduce and explain the full words when mentioning abbreviations for the first time e.g. NGOs.

The authors should explain the tools used for reporting and acquiring complaints. It is hard to compare manuscripts dealing with letters and those with systematic questionnaires. It should be grouped differently.

The authors should explain the context of complaints. Do the complaints come from people insured through state insurance or private?

The authors should state the limitations of this research and methodology and how they acquired and chose these exact studies.

Comments on the Quality of English Language

The manuscript needs minor corrections regarding spelling and grammar.

Author Response

We thank you for your comments and respond your observations as follows:

This manuscript is very well written, yet it needs some minor changes to be done prior to publishing.

Given that the editor has given us only 10 days to send back a reviewed version, perhaps it may be possible to edit the text after it is accepted.

The abstract should contain an aim as well as a conclusion.

The paper's aim, which has been contrasted in the abstract, addresses the explanation for the difference in the kinds of complaints observed in Brazil compared to other countries. The abstract concludes by suggesting that a family logic is not noticeable, because patients may be afraid to point out medical issues. At the same time, the medical professional logic seems to stand out due to its power to regulate and exercise its practices.

The authors should omit referencing in the abstract section.

It reads now - complaint categorization developed by Reader and colleagues

Also, at the end of the introduction, the authors should state the aim of this research.

We understand that it was not clear where the introduction ended. We’ve modified to make it clearer. The text reads – (the study) explores the possible explanations for these (complaint) differences based on the institutional logics that the actors privilege and how they may play out in the healthcare field.

The keywords should be according to the PubMed Mesh subheadings.

The first keyword is 'complaints' because it is what the study is about, which does not appear in the list; the second, instead of healthcare we can use comprehensive healthcare, as it appears in the subheadings; institutional logics is the theory used in the study's analysis and it is mentioned as a subheading; we are changing customer satisfaction for consumer satisfaction, the equivalent subheading.

The authors should introduce and explain the full words when mentioning abbreviations for the first time e.g. NGOs.

 Done.

The authors should explain the tools used for reporting and acquiring complaints. It is hard to compare manuscripts dealing with letters and those with systematic questionnaires. It should be grouped differently.

 We removed the letters mention. However, our study does not report complaint surveys, but the analysis made by other studies about their characteristics.

The authors should explain the context of complaints. Do the complaints come from people insured through state insurance or private?

 In the studies other than Brazilian, most of the studies mention health systems that are managed by the state. Nonetheless, the nature of the study has to do with the kinds of complaints that are addressed for judgment and the reasons it is substantially different in Brazil.

The authors should state the limitations of this research and methodology and how they acquired and chose these exact studies.

We added to section 2, method, how the studies were chosen. We also mention the study’s limitations at the end.

Reviewer 3 Report

Comments and Suggestions for Authors

The article is well-written in terms of content. It can be said that the article has a review character since the authors rely on already published studies. In any case, the article can contribute to the literature. To improve the quality of the text, it is worth considering a few suggestions/comments:

Line 119 – The authors mention "the right to dignity." It is not clear how this criterion should be understood. In healthcare, it is generally referred to as the respect for dignity. In other contexts (e.g., in law), dignity is considered the foundation of human rights. Perhaps the authors mean the right to equal and dignified treatment?

Lines 136-139 – The data on complaints in China should be described in more detail. For example, it would be worth mentioning how many complaints were analyzed.

General remark (This note pertains to the section "Healthcare complaining in different countries"): When describing studies from different countries, the authors sometimes refer to the number of complaints analyzed, while at other times, they do not. This lack of consistency is noticeable. It would be beneficial to standardize the data presented: either introduce the number of complaints analyzed for each country or omit the number and focus on specific criteria.

The article lacks a "Limitations" section. The authors do not mention the limitations of their study. It should be emphasized that every study has its limitations. This article is no exception. This section should be added and completed as it is a methodological requirement.

Author Response

We thank you for your comments and respond your observations as follows:

The article is well-written in terms of content. It can be said that the article has a review character since the authors rely on already published studies. In any case, the article can contribute to the literature. To improve the quality of the text, it is worth considering a few suggestions/comments:

Line 119 – The authors mention "the right to dignity." It is not clear how this criterion should be understood. In healthcare, it is generally referred to as the respect for dignity. In other contexts (e.g., in law), dignity is considered the foundation of human rights. Perhaps the authors mean the right to equal and dignified treatment?

Corrected.

Lines 136-139 – The data on complaints in China should be described in more detail. For example, it would be worth mentioning how many complaints were analyzed.

Following the suggestion below, we have removed complaint figures from the text.

General remark (This note pertains to the section "Healthcare complaining in different countries"): When describing studies from different countries, the authors sometimes refer to the number of complaints analyzed, while at other times, they do not. This lack of consistency is noticeable. It would be beneficial to standardize the data presented: either introduce the number of complaints analyzed for each country or omit the number and focus on specific criteria.

Following this suggestion, we have removed complaint figures from the text.

The article lacks a "Limitations" section. The authors do not mention the limitations of their study. It should be emphasized that every study has its limitations. This article is no exception. This section should be added and completed as it is a methodological requirement.

We have added a limitations explanation explicitly at the end.

Round 2

Reviewer 1 Report

Comments and Suggestions for Authors

Dear Author,
Thank you for the additions.

I acknowledge that you have seacrh literature and read numerous studies. However, your methodology and descriptions are not systematically executed/described enough to qualify as a systematic review. Hence my comment that it does not contribute significantly to the field.

Nevertheless, I truly recognize your considerable effort and believe it is relevant to explain the differences in complaint patterns that you highlight.

Reviewer 3 Report

Comments and Suggestions for Authors

I have no more comments.